# FDA-Approved Drug Screening for Compounds That Facilitate Hematopoietic Stem and Progenitor Cells (HSPCs) Expansion in Zebrafish

**DOI:** 10.3390/cells10082149

**Published:** 2021-08-20

**Authors:** Zhi Feng, Chenyu Lin, Limei Tu, Ming Su, Chunyu Song, Shengnan Liu, Michael Edbert Suryanto, Chung-Der Hsiao, Li Li

**Affiliations:** 1Key Laboratory of Freshwater Fish Reproduction and Development, Institute of Developmental Biology and Regenerative Medicine, Ministry of Education, Southwest University, Chongqing 400715, China; jh251257813@email.swu.edu.cn (Z.F.); a18090083773@email.swu.edu.cn (C.L.); tlm253@email.swu.edu.cn (L.T.); suming1112@email.swu.edu.cn (M.S.); scyaidyj@email.swu.edu.cn (C.S.); ss845523483@email.swu.edu.cn (S.L.); 2Research Center of Stem Cells and Ageing, Chongqing Institute of Green and Intelligent Technology, Chinese Academy of Sciences, Chongqing 400714, China; 3Department of Bioscience Technology, Chung Yuan Christian University, Taoyuan 320314, Taiwan; michael.edbert93@gmail.com; 4Center for Nanotechnology, Chung Yuan Christian University, Taoyuan 320314, Taiwan; 5Research Center for Aquatic Toxicology and Pharmacology, Chung Yuan Christian University, Taoyuan 320314, Taiwan

**Keywords:** HSPCs expansion, drug screening, zebrafish, vitamins

## Abstract

Hematopoietic stem cells (HSCs) are a specialized subset of cells with self-renewal and multilineage differentiation potency, which are essential for their function in bone marrow or umbilical cord blood transplantation to treat blood disorders. Expanding the hematopoietic stem and progenitor cells (HSPCs) ex vivo is essential to understand the HSPCs-based therapies potency. Here, we established a screening system in zebrafish by adopting an FDA-approved drug library to identify candidates that could facilitate HSPC expansion. To date, we have screened 171 drugs of 7 categories, including antibacterial, antineoplastic, glucocorticoid, NSAIDS, vitamins, antidepressant, and antipsychotic drugs. We found 21 drugs that contributed to HSPCs expansion, 32 drugs’ administration caused HSPCs diminishment and 118 drugs’ treatment elicited no effect on HSPCs amplification. Among these drugs, we further investigated the vitamin drugs ergocalciferol and panthenol, taking advantage of their acceptability, limited side-effects, and easy delivery. These two drugs, in particular, efficiently expanded the HSPCs pool in a dose-dependent manner. Their application even mitigated the compromised hematopoiesis in an *ikzf1^−/−^* mutant. Taken together, our study implied that the larval zebrafish is a suitable model for drug repurposing of effective molecules (especially those already approved for clinical use) that can facilitate HSPCs expansion.

## 1. Introduction

Hematopoietic stem and progenitor cells (HSPCs) transplantation has been a major stem cell-based curative therapy in the treatment of hematologic diseases, including leukemia, immune deficiencies, hemoglobinopathies, and metabolism-based disorders, since the late 1950s, due to their capacity of reconstructing blood system [1]. Accessibility of bone marrow transplantation for patients is restricted by the short availability of immune-matched donors [2]. Umbilical cord blood (UCB) has become an increasingly popular source of transplantable HSPCs because of its rapid availability with less-stringent immune-matching requirements [3]. Therefore, the ability to expand sufficient HSPCs prior to transplantation has great clinical significance.

HSPCs expansion is an extremely complicated process [4]. A complex extrinsic and intrinsic cell signaling network is required, involving Notch signaling [5], growth factors [6], and epigenetic modification [7]. Over the last decades, the amplification of HSPCs has been accomplished by an integrated array of divergent approaches, including optimization of cytokine cocktails, coculture systems, small molecules, and delivery systems for HSPCs expansion genes [8]. Consistently, extensive efforts have been put into finding culture conditions that sustain HSCs ex vivo expansion, spurring the improvements in transplantation treatment outcomes through the development of various cellular therapies [9].

Our research emphasis is in identifying small molecule drugs to be used as efficient agonists for HSPCs expansion. It was first reported that histone deacetylase (HDAC) inhibitor trichostatin A (TSA) [10] and histone methyltransferase inhibitor 5-aza-2-deoxycytidine (5azaD) [11] promoted the expansion of cord blood severe combined immunodeficient (SCID)-repopulating cells. Consistently, a variety of natural and synthetic molecules were also reported that could enhance the homing efficiency and promote engraftment of HSPCs with the bone marrow [12]. Because of the positive attributes of drug repurposing, we adopted an FDA-approved drug library to identify candidates in promoting HSPCs expansion in zebrafish larvae. Traditional drug discovery, including preclinical testing, phase I-III trials, and FDA approval, requires 12–16 years and costs 1–2 billion dollars. There is a growing interest in repurposing ‘old’ drugs to treat both common and rare diseases because it has less risks and the potential to reduce overall development costs and timelines [13,14].

The zebrafish (*Danio rerio*) has unique advantages, such as its small size, high reproduction rate, and transparency, which make it an ideal model organism to study hematopoiesis [15]. There are two distinct waves in zebrafish hematopoiesis, similar to that in higher vertebrate organisms. The primitive hematopoiesis produces myeloid cells and erythrocytes, while definitive hematopoiesis generates HSPCs [16]. Although zebrafish and mammals possess different main hematopoietic sites, both species develop major blood cell types that share common hematopoietic origins [17]. These features have allowed studies on zebrafish blood development to be applied in mammalian systems. Additionally, fish and mammals share a number of genes, signaling pathways, notable transcription factors that influence blood cell development, as well as the differentiation of HSPCs related to hematopoiesis [18]. For example, *runx1* marks HSPCs in both mice and fish. In differentiated populations, *gata1* regulates the erythroid lineage, while *p**u.1* and *c/ebpα* regulate the myeloid lineage, and *ikzf1* labels the lymphoid population [19].

Over the past two decades, a panel of experiments have been performed on drug exploration in zebrafish [20,21,22,23]. Among many great achievements, the most famous one was prostaglandin E2 (PGE2), which directly stimulated HSC production and engraftment. Recently, the phase II clinical trial of the PGE2 treatment was conducted on more than 150 patients who have received cord blood or mobilized peripheral blood stem cells treated with dmPGE2 [24,25]. In addition, the assessment of trifluoperazine to treat Diamond-Blackfan anemia was in the phase I clinical stage [26]. These findings indicate that zebrafish is a suitable model for large-scale drug screening to treat blood diseases.

In this study, we aimed to identify the effective candidates that can facilitate HSPCs expansion from an FDA-approved drug library. We selected 171 compounds, divided into 7 groups, to treat zebrafish at 3 days post-fertilization (dpf) and observed HSPCs variation by quantifying the pixels of *cmyb*^+^ signals in the CHT region at 4 dpf. By preliminary screening, we identified 21 drugs that could stimulate HSPCs proliferation and 32 drugs that diminished HSPCs. Among these 21drugs, we focused on 6 vitamin drugs with limited side-effects and easy delivery, without ruling out that the other 15 drugs also had potential functions for therapeutical use in HSPCs expansion ex vivo.

In our follow-up studies, we look deeper into these vitamin drugs’ effect on HSPCs expansion. Interestingly, ergocalciferol and panthenol showed the most significant effects. They promoted HSPCs expansion in a dosage-dependent manner and amplified hematopoiesis. Furthermore, when we used these drugs to treat *ikzf1^−/−^* mutants, which harbor compromised proliferation of HSPCs, they rescued the HSPCs expansion-defective phenotypes. These results indicated that ergocalciferol and panthenol had the potential for clinical application in HSPCs expansion and enrichment.

## 2. Materials and Methods

### 2.1. Zebrafish Maintenance

The wild type zebrafish (*Danio rerio*) line was purchased from China Zebrafish Resource Center (CZRC, China). The *Tg(CD41:GFP)* transgenic line [27] was used to label HSPCs and thrombocytes. They were raised and maintained according to a standard procedure. During the experimental period, pH volume ranged from 7.8 to 7.9. Dissolved oxygen ranged from 6.95 to 7.23 mg/L. Water temperature ranged from 26.5 to 28 °C. Concentrations of ammonia-N and nitrite nitrogen were maintained lower than 0.2 and 0.005 mg/L, respectively. Salinity of water was 0.2 ppt. Zebrafish were maintained in a 12:12 h light–dark cycle. Embryos were collected from natural spawning and raised at 28.5 °C in egg water with 0.003% 1-phenyl-2-thiourea (PTU) at 12 hpf. The maintenance procedures and experiments of zebrafish were complied with guidelines approved by the Ethics Committee of the College of Life Science, Southwest University (Chongqing, China) with Approval ID: 2,018,092,308. This guidance ensured a clean and disease-free comfortable living environment for the animals.

### 2.2. Drug Treatment

The FDA-approved drug library was purchased from MicroSource Discovery System (CT 06755-1500). Cholecalciferol (S4063), Calcitriol (S1466), and Calcifediol (S1469) were purchased from Selleck Chemicals (Houston, TX, USA). D-Pantothenicacid (B2002) was purchased from Apexbio (Boston, TX, USA). All these drugs were prepared as stock solutions by dissolving it in dimethyl sulfoxide (DMSO). For the treatments, the stock solution was diluted in egg water until reaching the working concentrations (5–20 µM). N-Phenylthiourea (PTU) was ordered from Sigma–Aldrich (St. Louis, MO, USA) and dissolved in water as a stock solution. Then, the PTU stock solution was diluted with egg water until reaching the desired working concentration (0.2 mM).

### 2.3. Whole-Mount In Situ Hybridization (WISH) and Quantification

A standard protocol [28] was followed for preparing antisense RNA probes. The following antisense probes labeled with digoxigenin were used: *cmyb*, *lyz,* and *gata1*. The embryos were fixed in 4% paraformaldehyde (PFA) at RT for 4 h. The signals were observed under a SteREO Discovery.V20 microscope (Carl Zeiss, Oberkochen, Baden-Wurttemberg, Germany). WISH signals were measured as previously described [29]. For *cmyb*^+^, *lyz*^+^, and *gata1*^+^ signals’ quantification, we selected the CHT region to estimate the signals areas (pixels) by using ImageJ (Rawak Software Inc., Stuttgart, Baden-Wurttemberg, Germany), as described previously [30].

### 2.4. Fluorescence Immunohistochemistry Staining

The larvae of zebrafish were stained with whole-mount fluorescence immunohistochemistry, as described in a previous study [31]. At the appropriate developmental stages, *Tg(CD41:GFP)* embryos were fixed with 4% paraformaldehyde. After the embryos were fixed, they were incubated with primary antibodies (Abs) (4 °C, overnight). Primary Abs against GFP (1:400, ab6658; Abcam, Cambridge, UK) and Phosphorylated histone 3 (1:400, sc-374669; Santa Cruz, Dallas, Texas, USA) were used. Secondary Abs used in the study included donkey anti-goat IgG Alexa Fluor 488 (1:400, A-11055; Invitrogen, Carlsbad, CA, USA) and donkey anti-rabbit IgG Alexa Fluor 647 (1:400, A-21447; Invitrogen) at 4˚C overnight. Finally, these embryos were mounted in 1% low-melting-point agarose and observed using the LSM700/880 confocal microscope (Carl Zeiss).

### 2.5. TUNEL Assay and EdU Incorporation

For TUNEL (Terminal deoxynucleotidyl transferase dUTP nick end labeling assay) assay and EdU cell proliferation labeling, the larvae were fixed in 4% paraformaldehyde, then stored in PBS solution at 4 ˚C overnight. As part of the TUNEL assay, detection and quantification of cell death were checked using In Situ Cell Death Detection Kit, TMR Red (12156792910; Roche, Basel, Switzerland), which was performed according to the manufacturer’s instructions. For EdU labeling, we injected the EdU (1 nL, 10 mM) into the heart of larvae [32] and fixed the sample after 2 h; the subsequent experiments followed the protocol of Click-iT EdU Kit (C10340; Life Technologies, Carlsbad, CA, USA) to label the proliferation cells at S phase.

### 2.6. Fluorescence-Activated Cell Sorting (FACS) Analysis

Single-cell suspensions of zebrafish CHT (caudal hematopoietic tissue) were fulfilled as reported previously [33]. In brief, 25 *Tg(CD41:GFP)* larvae CHT region were digested with 0.25% trypsin at 28.5 °C for 40 min. Then, the cell suspensions were obtained by pipetting and filtration of a 40 μm cell strainer. All FACS analyses were performed by MoFlo XDP (Beckman Coulter, Brea, CA, USA) following the manufacturer’s instructions and reported previously [34].

### 2.7. Quantification and Statistics

Statistical analyses were performed by GraphPad Prism6.0 (GraphPad Software Inc., San Diego, CA, USA). The positive signals areas in larval CHT (caudal hematopoietic tissue) were manually scored and double-confirmed blindly. All quantified data (Mean ± SEM) were analyzed by two-tailed Student’s *t* test. The significant difference was indicated by a *p*-value < 0.05 statistically.

## 3. Results

### 3.1. A Wide-Range Drug Screen for HSPCs Expansion in Zebrafish Using an FDA-Approved Library

During zebrafish hematopoiesis, the caudal hematopoietic tissue (CHT) region is functionally similar to the mammalian fetal liver, which is an HSPCs expansion site [35]. Therefore, we focused on this region and characterized the HSPCs proliferation signatures in different stages. We selected the *Tg(CD41:GFP)* transgenic line, in which GFP^high^ cells mark thrombocytes, while GFP^low^ cells label HSPCs [27]. Then, we used phosphorylated histone H3 (PH3) immunofluorescent staining to indicate proliferative cells in the G2/M phase (Figure 1A). The quantification data (Figure 1B) showed that PH3^+^ signals in CD41-GFP^low^ populations increased from 2.33% ± 0.55% (2.5 dpf, days post fertilization) to 9.96% ± 0.90% (4 dpf) and then decreased to 6.78% ± 0.89% (5 dpf).

Based on the statistical analysis, we concluded that HSPCs have high proliferation capacity from 3 to 4 dpf during zebrafish embryogenesis. Therefore, we paid attention to this time frame to design a preliminary drug screening system. Firstly, we collected 3 dpf wild-type embryos into 12-well plates prior to adding 1 mL egg water with 10 μM drug. After 24 h of treatment, we fixed these embryos at 4 dpf to detect the HSPCs by examining *cmyb* (a HSPCs marker) [36] signals using whole-mount in situ hybridization (Figure 1C). In subsequent experiments, we used this model to conduct large-scale drug screening to find candidates in promoting HSPCs expansion (results are summarized in Table A1 in Appendix B).

### 3.2. Identification and Characterization of Drugs in Controlling HSPCs Homeostasis

Based on the initial screening data, we divided the treated samples into normal (standard), increased, and decreased groups, according to the quantified areas of *cmyb**^+^* signals (Figure 2A and Table A1). Ultimately, we screened 171 FDA-approved drugs. Among them, a high percentage of compounds (69%, 118 of 171) failed to alter HSPC homeostasis, which was set as normal or standard. However, 21 (12%) and 32 (19%) drugs led to increased or decreased pools of *cmyb**^+^* HSPCs, respectively (Figure 2B). These drugs were classified to 7 groups, including antibacterial, antineoplastic, glucocorticoid, NSAIDS, vitamin, antidepressant, and antipsychotic drugs (Figure 2C).

We then focused on the drugs related to the expansion of *cmyb^+^* cells after application. The statistical results indicated that 10 molecules of antibacterial drugs (Figure 2D), 2 of antineoplastic drugs (Figure 2E), 6 of vitamin drugs (Figure 2F), and 3 of antidepressant/psychotic drugs (Figure 2G) promoted HSPCs (*cmyb^+^* signals areas) expansion from approximate 16,680 ± 608 pixels in the DMSO group to more than 27,290 ± 647 pixels in the treatment groups. Meanwhile, we also found that 8 antibacterial (Figure 2H), 8 antineoplastic (Figure 2I), 7 glucocorticoid (Figure 2J), 5 NSAIDS/vitamin (Figure 2K), and 4 antidepressant/psychotic drugs (Figure 2L) led to an obvious reduction of HSPCs areas to nearly 238 ± 107 pixels. Collectively, we found 21 potential compounds from an FDA-approved drug library that facilitated the HSPCs expansion in zebrafish embryos.

### 3.3. The Contribution of Vitamin Drugs to HSPCs Expansion and Mitigation of the Hematopoietic Phenotypes in Ikzf1^−/−^ Mutants by Ergocalciferol and Panthenol

Among the drugs that led to HSPCs expansion, we attempted to find suitable molecules for clinical application with limited side-effects and easy delivery. After deliberating, we determined that vitamin drugs met our requirement. Consistently, there is research demonstrating that vitamin D receptor (VDR) signaling is essential in HSPCs production and differentiation [37,38,39] and that vitamin A-retinoic acid signaling regulates HSC dormancy [40], suggesting that vitamin drugs may play an important role in HSPCs maintenance.

Our results indicated that biotin, α-tocopheryl acetate, ergocalciferol, panthenol, ascorbic acid, and retinol treatment led to an alteration in the HSPCs pool, compared to the DMSO group (Figure 3A). The statistical data revealed that ergocalciferol and panthenol treatment manifested a better effect on HSPCs expansion than other counterparts of vitamin drugs. They enlarged the areas of HSPCs from 17,530 ± 552 pixels to 26,490 ± 1084 pixels, compared to other compounds (Figure 2F). In order to validate the reasons behind the expansion after application of these vitamin drugs, we performed an EdU incorporation assay (Figure 3B). We quantified HSPCs populations (CD41-GFP ^low^ cells) and EdU^+^/CD41-GFP ^low^ ratio in the CHT region; the statistical data showed that the vitamin treatment led to an obvious augment of HSPCs populations and EdU^+^/CD41-GFP^low^ ratio (Figure 3C,D), especially ergocalciferol (131 ± 6; 91.88% ± 1.37%), panthenol (125 ± 4; 92.00% ± 1.66%), ascorbic acid (126 ± 4; 89.75% ± 2.22%), and retinol (130 ± 4; 85.25 ± 2.29), compared to DMSO (59 ± 5; 43.25% ± 2.54%). The results demonstrate that these drugs contribute remarkably to the HSPCs proliferation.

In order to validate the effects of vitamin drugs on HSPCs expansion, we set out to seek a HSPCs proliferation-defective mutant. Ikzf1 is a Krüppel-like zinc–finger transcription factor that plays a crucial role in the development of T and B cells. Additionally, loss of Ikzf1 leads to compromised HSPCs expansion [41]. Therefore, we used these six vitamin drugs to treat *ikzf1**^−/−^* mutants. The *cmyb* in situ hybridization results presented that only ergocalciferol and panthenol treatment enlarged the HSPCs population in *ikzf1**^−/−^* mutants (Figure 3E). The statistical results indicated that the rescue efficiency on the *ikzf1**^−/−^* mutants blood defect phenotypes was 50% (6/12) by ergocalciferol and 71.4% (10/14) by panthenol (Figure 3E). Consistently, we also adopted flow cytometry to analyze the proportion of CD41-GFP^low^ cells within the whole CHT cells after treating ergocalciferol and panthenol (Figure 3F). The quantification results indicated that, compared to the DMSO group (0.44% ± 0.041%), the proportion increased markedly to 0.78% ± 0.015% by ergocalciferol and 0.84% ± 0.019% by panthenol (Figure 3G). This data supports the drastic effects of ergocalciferol and panthenol on the HSPCs expansion.

### 3.4. The Dose-Dependent Effects of Ergocalciferol and Panthenol on HSPCs Expansion

Because of their impressive effect on HSPCs expansion and *ikzf1**^−^**^/^**^−^* mutants phenotypes mitigation, we selected ergocalciferol and panthenol for further study. Due to their parallel drug impact on HSPCs expansion, we were curious about whether the two molecules shared the similar structures. We referred to the structural formula of ergocalciferol (Appendix A) and panthenol (Appendix A). Ergocalciferol (C_28_H_44_O) and panthenol (C_9_H_19_NO_4_) belong to the vitamin D or vitamin B family, and there is an enormous range in molecular weight (396.65 to 205.25). Those data showed that ergocalciferol and panthenol structures were quite distinct and mechanisms on HSPCs expansion may be different. Furthermore, in order to elicit the impact of the two drugs on cell apoptosis, we counted TUNEL^+^ signals in the CHT region. The confocal images and statistical analysis indicated that ergocalciferol (5 ± 1) and panthenol (3 ± 1) failed to affect HSPCs apoptosis, compared to the DMSO group (3 ± 1) (Appendix A).

Since ergocalciferol and panthenol showed a promotion of HSPCs expansion, we detected the impacts of different concentrations of treatment, ranging from 5 to 20 μM (Figure 4A). Consistently, we observed increasing areas of cmyb+ HSPCs from low (5 μM) to high (20 μM) concentrations of the two drug treatments (Figure 4B), which indicated that ergocalciferol and panthenol regulated the HSPCs expansion in a dosage-dependent manner. HSPCs produce specific blood cells by a process of hematopoiesis. Therefore, we investigated whether the development of various blood lineages, such as granulocytes (lyz+) [42] and erythrocytes (gata1+) [43], could be promoted (Figure 4C). The quantification results indicated a significant enlargement of lyz+ and gata1+ signals areas after treatment with ergocalciferol (8678 ± 617 pixels; 16,350 ± 1816 pixels) and panthenol (7524 ± 468 pixels; 13,790 ± 1252 pixels), compared to DMSO (4980 ± 467 pixels; 8525 ± 768 pixels) (Figure 4D,E).

### 3.5. Comparison of the Analogs Effects of Ergocalciferol and Panthenol

Ergocalciferol is vitamin D2, and its analogs are cholecalciferol (vitamin D3), calcifediol (25-hydroxyvitamin D3), and calcitriol (1,25-dihydroxyvitamin D3). We counted *cmyb^+^* signals areas after treatment with these molecules (20 μM) at 3 dpf. Compared to the DMSO group (16,090 ± 659 pixels), calcitriol led to higher lethality. However, cholecalciferol (15,840 ± 821 pixels) and calcifediol (15,880 ± 485 pixels) had no notable effect on HSPCs expansion (Figure 5A,B). Panthenol is provitamin B5, and its analog is pantothenic acid (16,790 ± 689 pixels). It limitedly affected HSPCs expansion (Figure 5A,B). These results demonstrate that only ergocalciferol (29,110 ± 1399 pixels) and panthenol (29,340 ± 1014 pixels) are endowed with the capability to promote HSPCs expansion, and further highlight their significant potential in clinical applications.

## 4. Discussion

In the FDA approval process, 73–82% of projects remained active in Phase II; however, 57–60% of the projects failed because of poor efficacy due to insufficient of human data [44]. Therefore, screening FDA-approved ‘old’ drugs is advantageous because of their established safety testing in humans, which saves time and cost. To this end, we performed high-efficiency screening of FDA-approved compounds and attempted to identify candidates that promote HSPCs expansion—in total, we screened 171 drugs and obtained 21 drugs. However, our screening also identified 32 drugs that cause diminishment of HSPCs, which might be useful in the alleviation of malignant blood diseases like leukemia. Meanwhile, we discovered 118 drugs whose administration had no effects on HSPCs expansion, most likely because the concentration (10 μM) was not sufficient for drug efficiency display in our preliminary screening, which probably resulted in some useful drugs missing out. However, we had chosen the following concentration that was typically used for compound screening assays (1–10 μM) [45]. Nevertheless, we plan to increase drug concentration to screen these drugs again in future studies.

Among the 21 drugs, we focused on vitamin drugs because of their low side-effects. The conventional view is that vitamins are organic compounds that people need in small quantities from foods. Uptake deficiency leads to hypovitaminosis, such as nyctalopia (vitamin A deficiency) and rickets (vitamin D deficiency). Additionally, vitamins and their derivatives have been applied to clinical therapeutics, such as acute promyeloid leukemia (APL) [46,47]. From our results, we found six vitamins, including biotin, α-tocopheryl acetate, ergocalciferol, panthenol, ascorbic acid, and retinol, that contribute to HSPCs expansion to a large degree. Interestingly, ergocalciferol and panthenol were able to ameliorate the HSPCs expansion deficiency phenotypes in *ikzf1**^−^**^/^**^−^* mutants. Consistently, the impact on HSPC expansion was dose-dependent, indicating that the effect of the drugs is specific as it has a dose-dependent action.

As a steroid hormone, vitamin D plays a role in regulating the metabolism of calcium and phosphate. Ergocalciferol (vitamin D2) belongs to the vitamin D family and is derived from the plant sterol ergosterol [48]. To date, no study has uncovered its role in hematopoiesis. Its analog, 1,25(OH)D3, an active form of vitamin D3, has been reported to stimulate HSPCs production via vitamin D receptor (VDR)-induced transcription, which can activate the expression of inflammatory cytokine CXCL8. However, when we used this drug (calcitriol, 20 μM) to treat embryos at 3 dpf, it was found to be lethal to larval zebrafish. The probable reason was that zebrafish embryos were unbearable at this concentration. Therefore, we will attempt to find the appropriate concentrations for larval zebrafish, to investigate the effects in the future. Based on our results, other analogs made no contribution to HSPCs expansion. Therefore, we hypothesized that the modulation mechanism of ergocalciferol may depend on an undiscovered pathway. Panthenol (provitamin B5) is a precursor of pantothenic acid (vitamin B5), which is an essential part of coenzyme A. This enzyme plays a significant role in the metabolism of cells, including the transfer of the acyl group during fatty acid biosynthesis and gluconeogenesis. It also promotes fibroblast proliferation and therefore promotes wound healing [49,50]. Nonetheless, the role of panthenol in HSPCs expansion is unclear.

Although our emphasis was directed toward ergocalciferol and panthenol, it did not indicate that other drugs with promotion or inhibition activities were dispensable. For instance, drugs with inhibition activities may be used to treat diseases related to abnormal proliferation of blood cells. For a substantial portion of these drugs, the detailed mechanisms remain unclear. In the future, understanding and characterizing the specific cellular targets of these drugs will be important and interesting. In addition, we used a drug concentration of 10 μM in the preliminary screening to save time and improve efficiency. However, this concentration might be insufficient to achieve a medicinal effect for certain types of drugs due to the different efficacy possessed by each drug [51,52], which is highly related to its properties, binding receptor, mechanism, and signaling pathway.

In summary, we adopted zebrafish as an in vivo model system to screen and evaluate FDA-approved drugs. Our aim was to identify novel molecules that influence HSPCs proliferation and provide a basis to begin to explore possible drugs to facilitate HSPCs expansion. Ultimately, we validated the effectiveness of ergocalciferol and panthenol. In future studies, we will investigate the mechanism of drug action and explore the possibility of clinical trials.

## 5. Conclusions

Collectively, we conducted a wide-range screening of FDA-approved drugs and uncovered a series of compounds that stimulate HSPCs expansion in zebrafish embryos, especially ergocalciferol and panthenol. Our study demonstrates that these drugs have potential for clinical application. In the future, more studies are required to characterize these drug targets and determine their utility and efficacy in mammalian or human disease models.

## Figures and Tables

**Figure 1 cells-10-02149-f001:**
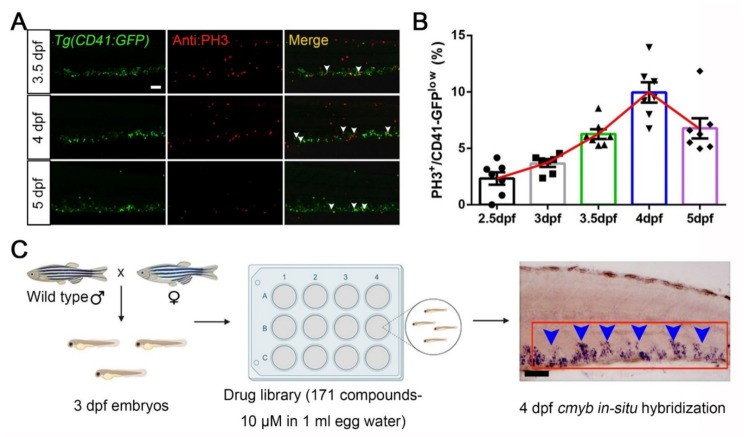
FDA-approved drug screening on compounds boosting HSPCs expansion in zebrafish. (**A**) The immunofluorescent staining images of *Tg(CD41-GFP)* and Phospho-Histone H3 (PH3). White arrowheads indicate merged signals. (**B**) Statistical diagram of corresponding percentage of PH3^+^ cells in CD41-GFP ^low^ populations (2.5 dpf, 2.33 ± 0.55; 3 dpf, 3.66 ± 0.31; 3.5 dpf, 6.26 ± 0.43; 4 dpf, 9.96 ± 0.90; 6.78 ± 0.90). (**C**) An overview of the experimental design in this study for drug screening by using zebrafish. A total of 20 wild-type embryos (3 dpf) were transferred to each well in a 12-well plate format. Then, embryos were administrated with one of 171 FDA-approved drugs for 24 h and screened for quantitative increases or decreases of signals in the CHT (caudal hematopoietic tissue) region at 4 dpf. The red box indicates the CHT region, and blue arrowheads indicate *cmyb^+^* signals. Mean ± SEM, *n* = 7; Scale bar, 50 μm.

**Figure 2 cells-10-02149-f002:**
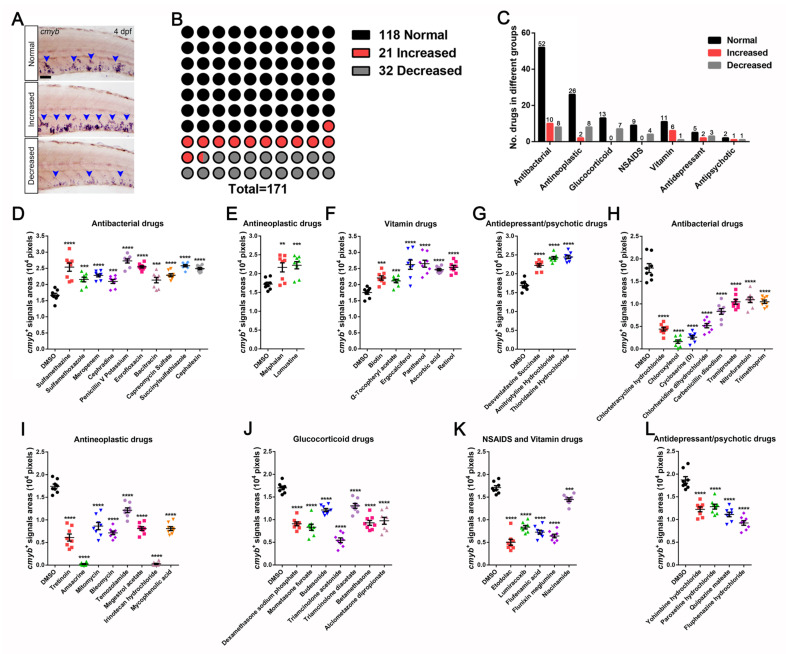
Preliminary FDA-approved drug screening results for *cmyb*^+^ HSPCs in zebrafish embryos. (**A**) Examples of screening and phenotyping between *cmyb* in situ hybridization phenotypes in comparison with the normal control zebrafish. (**B**) Statistical summary of drugs available, including normal signals (118 compounds), increased signals (21 compounds), and decreased signals (32 compounds). (**C**) Statistical diagram of drugs represented by each category. (**D**–**G**) The drug treatments augment *cmyb^+^* signals. (**D**) Antibacterial drugs (Pixels; DMSO, 16,680 ± 608; Sulfamethazine, 25,360 ± 1281; Sulfamethoxazole, 21,540 ± 699; Meropenem, 22,790 ± 443; Cephradine, 20,940 ± 616; Penicillin V Potassium, 27,290 ± 647; Enrofloxacin, 25,450 ± 355; Bacitracin, 21,350 ± 841; Capreomycin Sulfate, 22,860 ± 454; Succinylsulfathiazole, 25,800 ± 341; Cephalexin, 24,890 ± 348). (**E**) Antineoplastic drugs (Pixels; DMSO, 17,230 ± 509; Melphalan, 21,650 ± 1205; Lomustine, 22,230 ± 963). (**F**) Vitamin drugs (Pixels, DMSO, 17,530 ± 552; Biotin, 22,050 ± 651; α-Tochopheryl acetate, 21,130 ± 528; Ergocalciferol, 26,190 ± 1457; Panthenol, 26,490 ± 1084; Ascorbic acid, 24,610 ± 345; Retinol, 25,380 ± 622). (**G**) Antidepressant/psychotic drugs (Pixels; DMSO, 16,910 ± 483; Desvenlafaxine Succinate, 22,210 ± 437; Amitriptyline hydrochloride, 24,230 ± 362; Thioridazine hydrochloride, 24,450 ± 421). (**H**–**L**) The drug treatment diminishes *cmyb^+^* signals. (**H**) Antibacterial drugs (Pixels; DMSO, 17,940 ± 985; Chlortetracycline hydrochloride, 4400 ± 3402; Chloroxylenol, 1625 ± 380; Cycloserine (**D**), 2600 ± 369; Chlorhexidine dihydrochloride, 5163 ± 479; Carbenicillin disodium, 8325 ± 638; Tramiprosate, 10,450 ± 543; Nitrofurantoin, 10,900 ± 614; Trimethoprim, 10,480 ± 376). (**I**) Antineoplastic drugs (Pixels; DMSO, 17,410 ± 505; Tretinoin, 6088 ± 783; Amsacrine, 238 ± 107; Mitomycin, 8625 ± 802; Bleomycin, 7125 ± 394; Temozolamide, 12,130 ± 531; Megestrol acetate, 8075 ± 420; Irinotecan hydrochloride, 263 ± 145; Mycophenolic acid, 8063 ± 431). (**J**) Glucocorticoid drugs (Pixels; DMSO, 17,000 ± 398; Dexamethasone sodium phosphate, 9075 ± 433; Mometasone furoate, 8325 ± 676; Budesonide, 12,140 ± 332; Triamcinolone acetonide, 5463 ± 512; Triamcinolone diacetate, 13,040 ± 532; Betamethasone, 9288 ± 539; Alclometazone dipropionate, 9738 ± 767). (**K**) NSAIDS and Vitamin drugs (Pixels; DMSO, 17,080 ± 440; Etodolac, 5025 ± 663; Lumiracoxib, 8350 ± 385; Flufenamic acid, 7250 ± 436; Flunixin meglumine, 6400 ± 400; Niacinamide, 14,450 ± 383). (**L**) Antidepressant/psychotic drugs (Pixels; DMSO, 18,710 ± 729; Yohimbine hydrochloride, 12,240 ± 535; Paroxetine hydrochloride, 12,830 ± 546; Quipazine maleate, 11,100 ± 502; Fluphenazine hydrochloride, 9238 ± 509). Scale bar, 50 μm; Mean ± SEM, *n* = 8; ** *p*< 0.01, *** *p* < 0.001, **** *p* < 0.0001.

**Figure 3 cells-10-02149-f003:**
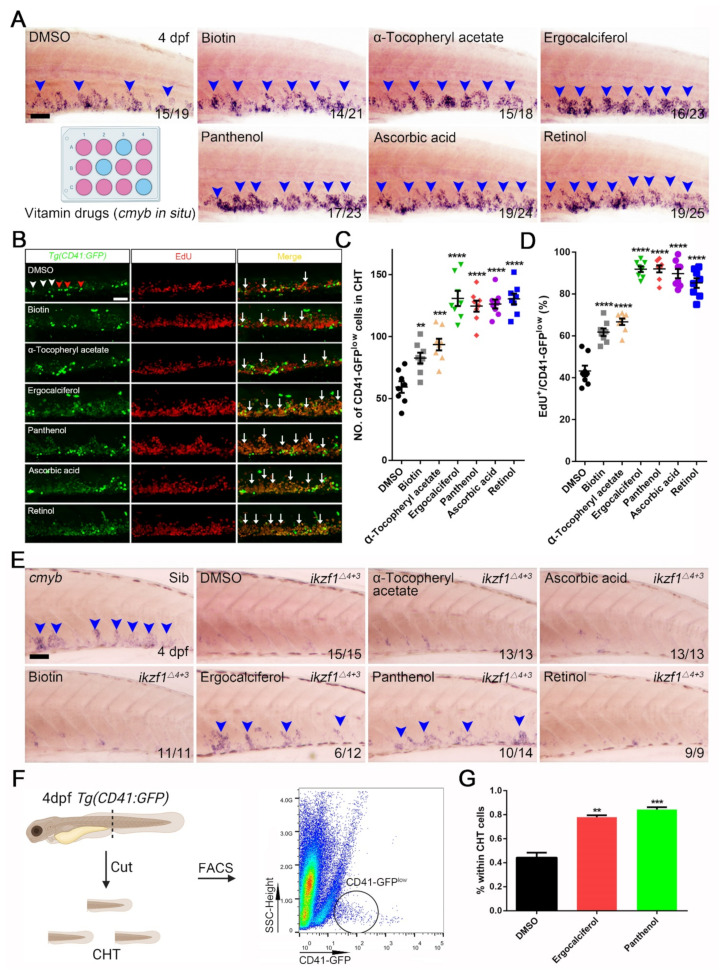
Vitamin drugs boost HSPCs expansion in zebrafish embryos and mitigate HSPCs expansion defective phenotype in *ikzf1**^−^**^/^**^−^* mutants. (**A**) Whole mount in situ hybridization (WISH) of *cmyb* after treating with vitamin drugs, including biotin, α-tocopheryl acetate, ergocalciferol, panthenol, ascorbic acid and retinol. Blue arrowheads indicate the *cmyb^+^* signals. (**B**) Double staining images of *Tg(CD41:GFP)* with EdU after treating with these vitamin drugs. White arrowheads indicate CD41-GFP ^low^ cells while red arrowheads indicate CD41-GFP ^high^ cells. The white arrows indicate the double labelled cells (EdU/CD41-GFP ^low^). (**C**) Quantification the number of CD41-GFP ^low^ cells. DMSO, 59 ± 5; Biotin, 83 ± 4; α-Tocopheryl acetate, 94 ± 5; Ergocalciferol, 131 ± 6; Panthenol, 125 ± 4; Ascorbic acid, 126 ± 4; Retinol, 130 ± 4). (**D**) Statistical result of EdU incorporation assay in *Tg(CD41:GFP)* (DMSO, 43.25 ± 2.54; Biotin, 61.75 ± 1.80; α-Tocopheryl acetate, 66.75 ± 1.56; Ergocalciferol, 91.88 ± 1.37; Panthenol, 92.00 ± 1.66; Ascorbic acid, 89.75 ± 2.22; Retinol, 85.25 ± 2.29). (**E**) WISH of *cmyb* after treating with α-Tocopheryl acetate, Ascorbic acid, Biotin, Ergocalciferol, Panthenol and Retinol. The blue arrowheads indicate *cmyb*^+^ signals in CHT region. (**F**) Schematic diagram of FACS analysis. The black circle indicates CD41-GFP^low^ population. (**G**) Quantification results of CD41-GFP^low^ cells within the whole CHT cells after ergocalciferol and panthenol treatment (Mean ± SEM, *n* = 3; DMSO, 0.44 ± 0.041; Ergocalciferol, 0.78 ± 0.015; Panthenol, 0.84 ± 0.019). Scale bar, 50 μm; ** *p* < 0.01, *** *p* < 0.001, **** *p* < 0.0001.

**Figure 4 cells-10-02149-f004:**
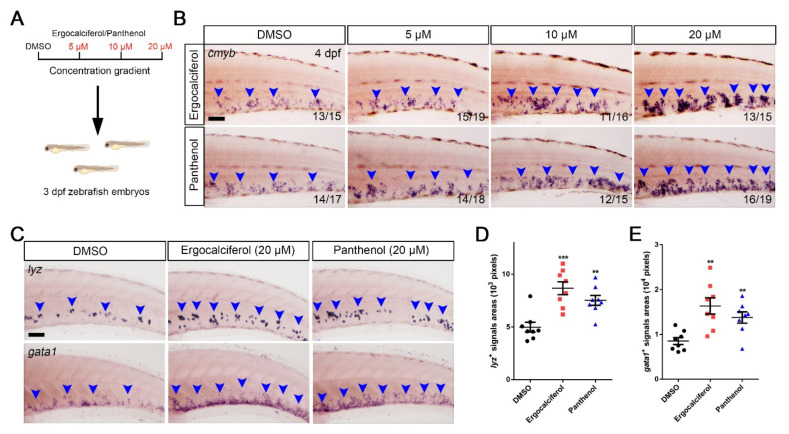
Ergocalciferol and panthenol facilitate HSPCs expansion in a dosage-dependent manner. (**A**) Schematic diagram of concentration gradient using ergocalciferol and panthenol. (**B**) WISH of *cmyb* with different concentration (5, 10, 20 μM). Blue arrowheads indicate *cmyb^+^* signals. (**C**) WISH of *lyz* (upper) and *gata1* (bottom) after treating with ergocalciferol and panthenol using 20 μM. (**D**,**E**) Quantification of *lyz^+^* (left) and *gata1*^+^ (right) signals areas in the CHT region. (Pixels, *lyz^+^* signals; DMSO, 4980 ± 467; Ergocalciferol, 8678 ± 617; Panthenol, 7524 ± 468; *gata1*^+^ signals; DMSO, 8525 ± 768; Ergocalciferol, 16,350 ± 1816; Panthenol, 13,790 ± 1252). Scale bar, 50 μm; Mean ± SEM, *n* = 8; ** *p* < 0.01, *** *p* <0.001.

**Figure 5 cells-10-02149-f005:**
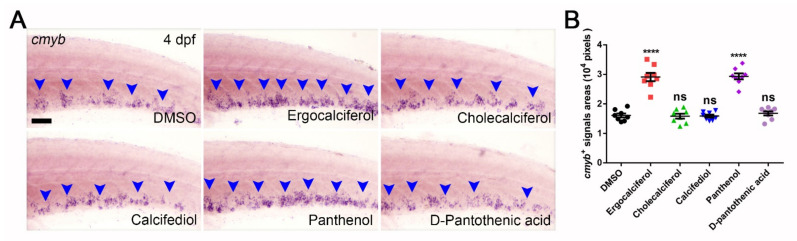
Comparison of the analog effects of ergocalciferol and panthenol. (**A**) WISH of *cmyb* after treating with ergocalciferol and its analogs (cholecalciferol and calcifediol) as well as panthenol and its analog (D-pantothenic acid). Blue arrowheads indicate *cmyb^+^* signals. (**B**) Statistical data of (**A**) (Pixels; DMSO, 16,090 ± 659; Ergocalciferol, 29,110 ± 1399; Cholecalciferol, 15,840 ± 821; Calcifediol, 15,880 ± 485; Panthenol, 29,340 ± 1014; D-pantothenic acid, 16,790 ± 689). Scale bar, 50 μm; Mean ± SEM, *n* = 8; ns, no significance; **** *p* < 0.0001.

## Data Availability

The original data presented in this study can be obtained from authors upon request.

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
