# Peer review of "FDA-Approved Drug Screening for Compounds That Facilitate Hematopoietic Stem and Progenitor Cells (HSPCs) Expansion in Zebrafish"

_cells, 2021, doi:10.3390/cells10082149_

Round 1

Reviewer 1 Report

Please consider the attached file

Author Response

Reviewer #1:

1.My major suggestion is related to the possibility to directly quantify the HSPCs population using the cmyb in situ staining. Although the cmyb marker is well assumed as a maker of HSPCs, the WISH technique is not precisely a quantitative analysis. When the Authors stated that observed a variation in the number of cmyb positive HSPCs, actually they were not specifically count for them. To quantify the signal of a WISH staining I would suggest to apply the quantification of the signal in all the experiments, by means of the use of pixel quantification in a selected area of the CHT. ImageJ or similar programs are useful for these quantification analyses.

Thank you very much for your kind suggestions! We apologize for adopting an inappropriate quantitative analysis to calculate WISH signals (cmyb+, lyz+ and gata1+ signals) in the previous manuscript. We agreed with your points and carefully re-quantified the WISH signals areas (Figure 1C, red box) by using ImageJ as suggested. The new data were included in Figure (2D-2L; 4H,I; 5B; Table A1) and the results were also pointed out in Section 3.2 (line 217-221), 3.4 (line 324-328), and 3.5 (line 344-349) accordingly. Thanks very much again!

  1. Moreover, I would suggest to replicate at least with the two selected more efficient vitamins ergocalciferol and panthenol the analyses in the CHT in the Tg(CD41:GFP) transgenic line. The use of this additional technique would enhance the potential of the system for the screening of compounds able to improve the HSPCs population, also allowing the direct quantification of this population with FACs analyses.

Thank you very much for your valuable advices. To more precisely evaluate the effects of vitamins on the HSPCs expansion and concerns the data as suggested, we additionally counted the number of CD41-GFP low cells (as an index for HSPCs population) after treating with the 6 vitamins (biotin, α-tocopheryl acetate, ergocalciferol, panthenol, ascorbic acid and retinol) (Text highlighted by yellow color on page 7, line 266-271). The quantification results indicated that all the six vitamins have the notable promotion roles, and ergocalciferol, panthenol, ascorbic acid and retinol presented more drastic effects among the family at the same treatment conditions. The quantification data was included in Figure 3C. Meanwhile, we adopted flow cytometry to analyze the proportion of CD41-GFPlow within the whole CHT cells as suggested, the related information and comment were included in the revised manuscript accordingly (Marked by yellow highlighted text on page 7, line 281-286). The quantification data of FACS analysis was included in Figure 3F-G. Thanks again!

3.Concerning the drug screening, among the 171 drugs of the FDA panel, 118 didn’t elicit any effect on HSPCs expansion neither increase or decrease. I was wondering if, by increasing the concentration of those drugs, they could promote HSPCs differentiation or reduction. This statement should be at least discussed in the manuscript.

Thank you very much for the wonderful suggestion and comments. We are sorry for the limited discussion on the possible dosage effects of these drugs on the HSPCs expansion. For the drug screening, some consideration should be made regarding the screening concentration. Actually, we followed compound screening assays that are typically run at 1 up to 10 µM [1]. Other study also discussed that there is potential risk of preclinical toxicity at an exposure level of >10 µM [2]. We considered and discussed this point as suggested in the revised manuscript accordingly (Marked by yellow highlighted color on page 11, line 366-370).

  1. Hughes, J.P., Rees, S., Kalindjian, S.B., & Philpott, K.L. Principles of early drug discovery. British journal of pharmacology, 2011. 162, 1239-1249.
  2. Van Vleet, T.R., Liguori, M.J., Lynch III, J.J., Rao, M., & Warder, S. Screening strategies and methods for better off-target liability prediction and identification of small-molecule pharmaceuticals. SLAS DISCOVERY: Advancing Life Sciences R&D, 2019. 24, 1-24.

  1. I was wondering why, among the 6 vitamins, the Authors choose to concentrate the further experiments only on ergocalciferol and panthenol. Indeed, from Figure 3C it seems that also ascorbic acid and retinol elicited the same effects. Moreover, also ascorbic acid and retinol are easy to deliver with few side effects. I would suggest to better discuss the choice.

Thank you very much for your kind comments. We selected ergocalciferol and panthenol for further investigation based on the data that these two molecules of six vitamin factors well mitigated the defective expansion phenotypes of cmyb+ HSPCs in ikzf-/- mutant embryos. We apologized for the improper description of this point in the previous manuscript, which caused a puzzle on the data presentation. We currently reorganized the data by changing previous Figure 5C to the new Figure 3E in the revised manuscript. We hope that this modification could explain and highlight the unique important signatures of ergocalciferol and panthenol in the vitamin family. The related information and discussion were included in the revised manuscript accordingly (Marked by yellow background on page 7, line 273-281).

  1. Another major point is related to the blood cells lineage derived from HSPCs. The graphs in Figure 4H didn’t show a clear effect of ergocalciferol and panthenol. Indeed, the expression of gata1 expression (erythrocyte lineage) was not different in the two categories nor the panthenol effect on granulocytes with panthenol administration. The HSPCs derived blood cell lineage must be investigated more into details to confirm the statements of the Authors or show other results (for example by increasing the concentration of the vitamins administrated).

We are quite sorry for the unclear presentation of the data in previous Figure 4H. As you suggested, we increased the volume of ergocalciferol and panthenol to 20µM and then carefully quantified the WISH signals of lyz+(granulocytes) and gata1+(erythrocytes) by pixels of areas after application. The data indicated that both chemicals drastically enlarged the populations of the lyz+ and gata1+cells. The newly WISH picture and quantification data were included in the revised Figure 4 G-L. And the description was added in the revised manuscript that was highlighted by yellow background on page 9 (line 324-328). Thank you again for your kind suggestions.

  1. About Figure 4, it is not clear why the Authors decided to compare the structure of ergocalciferol and panthenol and why they should elicit apoptotic effects. I agree with the Authors that the evaluation of an adverse effect (for example apoptosis induction), is of particular importance when proposing a drug therapy. However, since the Authors observed an increase of the HSPCs population with these two vitamins, why they should elicit apoptosis?

We are very sorry for the unclear elucidation of previous Figure 4A-D. The similarly intensive impact of ergocalciferol and panthenol on HSPCs expansion led us to explore whether the effects were possibly induced by their structure similarities. We therefore compared their structural formula and molecular weight. The result indicated that the two drugs were quite different in their structure, suggesting that mechanisms of both factors on HSPCs expansion were probably different. This point is an interesting issue for the future study. We included this information in the revised manuscript (Marked by yellow background on page 9). We conducted the TUNEL assay to estimate the apoptotic roles of two molecules on the HSPCs, because a panel study revealed that the reduced cell death was probably served as an additional reason for the enlargement of HSPCs pool [1-3].To exclude the apoptotic influence of two chemicals on the HSPCs and precisely confirm their pro-proliferative roles for the expansion, we did the TUNEL assay. We apologized for the puzzled explanation of the TUNEL and structure information in our previous presentation. The related information was improved and included in the revised manuscript accordingly (Marked by yellow background on page 9).

  1. Pelus, L.M., J. Hoggatt, and P. Singh, Pulse exposure of haematopoietic grafts to prostaglandin E2 in vitro facilitates engraftment and recovery.Cell Prolif, 2011. 44 Suppl 1, 22-9.
  2. Qiao, J., et al., Endothelial progenitor cells improve the quality of transplanted hematopoietic stem cells and maintain longer term effects in mice.Ann Hematol, 2017. 96, 107-114.
  3. Holmes, T., et al., Glycogen synthase kinase-3beta inhibition preserves hematopoietic stem cell activity and inhibits leukemic cell growth.Stem Cells, 2008. 26, 1288-97.

Minor revisions:

GENERAL

-Scale bar is lacking in all the WISH figures.

We are quite sorry for lacking the scale bars in all the WISH figures. In the revised manuscript, we added the scale bar (50 μm) in all WISH figures accordingly.

-Please carefully check for scale bar dimension. For example, the magnification of Figure 3B seems identical to Figure 4C; the scale bar dimension in figure legends are indicated as 20 microm but in the figure the scale bar dimension is different between Figure 3B and 4C.

We are sorry for the scale bar dimension in the previous images as mentioned. We carefully re-checked the scale bar dimension and altered them to 50 μm in the revised Figure 1A, 3B and 4C to provide a clearer presentation. We selected about 5-somites stage to present EdU signals in Figure 3B, while 3-somites stage to show TUNEL+ signals in Figure 4C. This is the reason of the different scale bar dimension between the two figures. The improved information was included in the revised manuscript. Thanks very much for your kind advices!

ABSTRACT

-Add also negative modulators among the drugs that exerted an effect on HSPCs differentiation.

Thank you for your kind suggestion. We added the information in abstract as suggested (Marked by yellow background on page 1, line 27-28).

-Last sentence: “Taken together, our study implied that the larval zebrafish is a suitable model to screen novel effective molecules to facilitate HSPCs expansion using ‘old’ drugs, which have the potential for clinical application”. I would suggest to modify as: “Taken together, our study implied that the larval zebrafish is a suitable model for drug repurposing of effective molecules, especially those already approved for clinical use, that can facilitate HSPCs expansion”.

Many thanks for your kind suggestion, and we appreciate the appropriate correction. We edited this sentence in the revised manuscript as suggested (Marked by yellow background on page 1, line 33-34).

INTRODUCTION

-“Our research emphasis is IN identifying…”

-“Consistently, it WAS also reported a variety of natural and synthetic molecules THAT may enhance the homing efficiency and promote engraftment of HSPCs with the bone marrow”.

-Move the last sentence of the paragraph immediately after the ref n. 12. “Because of the positive attributes of drug repurposing, we adopted an FDA-approved drug library to identify candidates in promoting HSPCs expansion IN ZEBRAFISH LARVAE”.

Thank you for your kind suggestion. In our revised manuscript, we corrected the corresponding parts as suggested on page 2 (line 55-63).

-Gene names should be in italics if zebrafish genes (i.e runx1, gata1a etc.)

Many thanks for your valuable correction, we italicized these gene names on page 2 as suggested in the revised manuscript.

-“In this study, we aimed to identify the effective candidates that can facilitate HSPCs expansion from an FDA-approved drug library. We selected 171 compounds, dividing into 7 groups, to treat zebrafish at 3 days post-fertilization (dpf) and observed HSPCs numbers by in situ hybridization 1 day later”. Please modify the sentence accordingly to my concern about the methodology for cmyb positive cell count.

Thanks very much for your valuable suggestion. In our revised manuscript, we improved this sentence as suggested (Marked by yellow background on page 2, line 92).

-“By preliminary screening, we identified 21 drugs that could stimulate HSPCs proliferation”. Add a comment about the drugs that reduce HSPCs proliferation

Many thanks to your kind suggestion. We are quite sorry for lacking the description about the drugs that reduce HSPCs proliferation. We added the comments in the revised manuscript accordingly (Marked by yellow background on page 2, line 93-94).

-“Among these drugs, we focused on 6 vitamin drugs with limited side effects and easy delivery”. It is conceivable that also other drugs might be suitable for therapeutical use. Please add a comment on this.

Thank you for your kind suggestion. We add the comment information in the revised manuscript accordingly (Marked by yellow background on page 2, line 95-96). Many thanks again!

Reviewer 2 Report

[Cells] Manuscript ID: cells-1306180 – Article

In this manuscript the well-established zebrafish larval model was used to perform a repurposing screen for previously approved drugs that increase HSPC proliferation between 3-4 dpf. A number of candidates across different classes were identified and the authors chose to focus on ergocalciferol and panthenol from the vitamin class. HSPC expansion was limited to these specific compounds and not their analogues. Whether the increase in HSPCs translated to a corresponding increase in mature haematopoietic lineages, and the ability to rescue the HSPC proliferation defect in the ikzf1 mutant were also investigated.

Comments: This is a nice study requiring some clarification on data interpretation. Specific comments are below.

2.3 Include a WISH standard protocol ref

2.4 Suggest replacing proper with “appropriate developmental stage”

2.5 EdU into larvae heart – ref

2.6 One-tailed T-test: Significance is reported for drugs that either increase or decrease HSPC proliferation. It would not have been known in which direction the drugs would alter HSPCs (up or down) at the outset of the screen, therefore a two-tailed T-test would have been more appropriate, and a stronger indicator of significance. Please justify use of one-tailed T test and/or re-run data analysis with two-tailed test.

3.3. “They enlarged the number of HSPCs…to more than 151”; the data are reported as 149 for ergocalciferol. Please correct text to reflect this.

Fig. 2K. Correct “Viatnim” to “Vitamin”

Fig. 3B. Suggest including arrowheads to distinguish CD41-hi thrombocytes from CD41-lo HSPCs, and to highlight double labelled cells (EdU/CD41-lo).

Fig. 4H. What is meant by “at different stages”? What is the threshold for “normal” versus “increased” signal groups? How are these distinguished? What is the n for each group in this figure? What were the results for the other concentrations? What test of significance was used to support the claim of “significant enlargement of lyz+ and gata1+ cells after treatment with ergocalciferol and panthenol”? Please clarify or to test the robustness of the claim, repeat twice more with all data presented.

Fig. 5D. What was the statistical test used? How are the medium and strong rescue categories defined? If n=8 for this experiment, then there are only 2 embryos in the medium and strong rescue categories for ergocalciferol. Please clarify or to test the robustness of the claim, repeat twice more with larger n numbers with all data presented.

Table A1: change “expend” to “expand”

Author Response

Reviewer #2:

-2.3 Include a WISH standard protocol ref.

Thank you for your kind suggestion, we apologized for the lack of reference about WISH standard protocol. We added the reference as suggested in our revised manuscription page 3 (line 136).

-2.4 Suggest replacing proper with “appropriate developmental stage”

Many thanks for your kind suggestion, we replaced proper with appropriate as suggested on page 3 (line 139).

-2.5 EdU into larvae heart – ref.

Thank you for your valuable advices, we are quite sorry for lacking the reference about injecting EdU into larvae heart. We added the reference as suggested in the revised manuscription page 4 (line 153).

-2.6 One-tailed T-test: Significance is reported for drugs that either increase or decrease HSPC proliferation. It would not have been known in which direction the drugs would alter HSPCs (up or down) at the outset of the screen, therefore a two-tailed T-test would have been more appropriate, and a stronger indicator of significance. Please justify use of one-tailed T test and/or re-run data analysis with two-tailed test.

Many thanks for your valuable advices. The authors agreed with the reviewer suggestion. Therefore, we re-run all the statistical data by using two-tailed T-test accordingly. Thank you very much again.

-3.3. “They enlarged the number of HSPCs…to more than 151”; the data are reported as 149 for ergocalciferol. Please correct text to reflect this.

Many thanks for your kind suggestion. In the revised manuscript, we adopted a more appropriate method to quantify WISH signals areas (pixels) by using ImageJ. And we modified this sentence to “They enlarged the areas of cmyb+ signals from 17530 ± 552 pixels to 26490 ± 1084 pixels” (Marked by yellow background on page 7, line 258).

-Fig. 2K. Correct “Viatnim” to “Vitamin”.

Thank you very much! The authors apologize about the spelling mistake, then correct the “Viatmin” to “Vitamin” in the revised manuscript.

-Fig. 3B. Suggest including arrowheads to distinguish CD41-hi thrombocytes from CD41-lo HSPCs, and to highlight double labelled cells (EdU/CD41-lo).

Many thanks for your kind suggestions. We used white arrowheads to indicate CD41-GFPlow cells, red arrowheads to indicate CD41-GFP high cells and white arrows to highlight the double labelled cells (EdU/CD41-GFP low) accordingly.

-Fig. 4H. What is meant by “at different stages”? What is the threshold for “normal” versus “increased” signal groups? How are these distinguished? What is the n for each group in this figure? What were the results for the other concentrations? What test of significance was used to support the claim of “significant enlargement of lyz+ and gata1+ cells after treatment with ergocalciferol and panthenol”? Please clarify or to test the robustness of the claim, repeat twice more with all data presented.

We are quite sorry for the unclear presentation on the previous Figure 4H, which was also concerned by reviewer 1. Based on the comments and suggestion from both of you, we increased the concentration of the molecules from 10 μM to 20 μM in the current manuscript. We repeated this experiment and adopted an appropriate method to quantify the WISH signals. The new data indicated a clear increment of lyz+ and gata1+ cells after treatment with ergocalciferol and panthenol. The revised figures were included in the present Figure 4G-I. The related description and information were added in the revised manuscript accordingly (Marked by yellow background on page 9, line 319-322).

-Fig. 5D. What was the statistical test used? How are the medium and strong rescue categories defined? If n=8 for this experiment, then there are only 2 embryos in the medium and strong rescue categories for ergocalciferol. Please clarify or to test the robustness of the claim, repeat twice more with larger n numbers with all data presented.

We apologize for the unclear description about Figure 5D. This point was also mentioned by Reviewer 1. We repeated this experiment and selected representative figures after treating with the 6 vitamins in the present Fig 3E. If a drug treatment mitigated the phenotype of reduced HSPCs in ikzf1-/-at 4 dpf, we believed that this candidate was effective. We marked the ratio of effective outcomes of the drug at the bottom right corner of each figure in the revised Figure 3E. For example, as for ergocalciferol treatment, the 6/12 in the right corner means that the phenotypes of 6 mutants were obviously rescued in the total of 12 ikzf1-/- mutant larvae. The corresponding description was included in the revised manuscript accordingly (Marked by yellow background on page 7, line 268-276).

-Table A1: change “expend” to “expand”

Many thanks for your valuable advices, and we apologize for the spelling mistake. We correct “expend” to “expand” in our revised manuscription page 13. Many thanks again for your kind suggestions and comments!

Round 2

Reviewer 1 Report

I thank the Authors that have addressed the majority of my concerns. However, minor revisions are still present and need to be addressed before final acceptance in CELLS.
MATERIALS AND METHODS
-Drug treatments: “While, N-Phenylthiourea (PTU) was ordered from Sigma-Aldrich (St. Louis USA) and dissolved in water as a stock solution”. Remove “While” at the beginning of the sentence.
RESULTS
-Reference for cmyb expression in HSPCs is lacking. Which probe?
-3.3 “Therefore, we used these 6 vitamin drugs to treat ikzf1-/-mutants”. ikzf1-/- in italic
-3.3 “The cmyb in situ hybridization” in situ in italic
-3.4 It is still not clear to me why the two drugs might have an effect on HSPCs apoptosis. Since the authors have demonstrated that the two drugs expand the HSPCs why they should test for apoptosis in this compartment? If the Authors want to demonstrate that the drugs are not toxic for the HSPCs, even if it is not expected as they expand and not reduce this population, they have to present the data differently. Otherwise, I would suggest to mitigate the strength of the data presenting them as supplementary figure.
-3.4 References for lyz and gata1a are lacking. Which probes?
DISCUSSION
-“Consistently, the impact on HSPC expansion was dose-dependent, indicating that ergocalciferol and panthenol have enormous potential for clinical application to induce HSPCs expansion”. This is not the conclusion of the results. The conclusion is that the effect of the drugs is specific as is has a dose-dependent action. This has not been addressed by the Authors in the revised version of the manuscript.
-“However, when we used this drug (calcitriol) to treat embryos at 3 dpf, it was lethal to larval zebrafish”. At which dose? Maybe reducing the dose of calcitrol the Authors could have less lethality and expansion of HSPCs? I would suggest to discuss about it in the revised version of the manuscript.
FIGURES
-Figure 2 legend in situ in italic
-Figure 3 I would suggest to put the control DMSO in the upper part of the figure replacing the scheme of the wells.
-Figure 3A in the legend in situ in italics
-Figure 4G ergocalcipherol in the box. The L is lacking
-Figure 5A is not clear I would suggest to remove it.

Author Response

Reviewer #1: Minor revision

MATERIALS AND METHODS

-Drug treatments: “While, N-Phenylthiourea (PTU) was ordered from Sigma-Aldrich (St. Louis USA) and dissolved in water as a stock solution”. Remove “While” at the beginning of the sentence.

Thank you for your kind suggestion. In our revised manuscript, we corrected the corresponding parts as suggested in page 3 (line 125).

RESULTS

-Reference for cmyb expression in HSPCs is lacking. Which probe?

Thank you for your kind suggestion, we apologized for the lack of reference about cmyb expression in HSPCs. We added the reference as suggested in our revised manuscript (page 4, line 184).

-3.3 “Therefore, we used these 6 vitamin drugs to treat ikzf1-/-mutants”. ikzf1-/- in italic

-3.3 “The cmyb in situ hybridization” in situ in italic

Many thanks for your valuable suggestions, we corrected the corresponding parts accordingly (page 7, line 272).

-3.4 It is still not clear to me why the two drugs might have an effect on HSPCs apoptosis. Since the authors have demonstrated that the two drugs expand the HSPCs why they should test for apoptosis in this compartment? If the Authors want to demonstrate that the drugs are not toxic for the HSPCs, even if it is not expected as they expand and not reduce this population, they have to present the data differently. Otherwise, I would suggest to mitigate the strength of the data presenting them as supplementary figure.

Many thanks for your valuable advices. In our revised manuscript, we reorganized the data by changing previous Figure 4A-D to new supplementary figure 1A-D as suggested. Thanks again!

-3.4 References for lyz and gata1a are lacking. Which probes?

Thanks for your valuable suggestions. In our revised manuscript, we added the references as suggested in page 9 (line 319).

DISCUSSION

-“Consistently, the impact on HSPC expansion was dose-dependent, indicating that ergocalciferol and panthenol have enormous potential for clinical application to induce HSPCs expansion”. This is not the conclusion of the results. The conclusion is that the effect of the drugs is specific as is has a dose-dependent action. This has not been addressed by the Authors in the revised version of the manuscript.

Many thanks for your valuable advices. We are quite sorry for unclear conclusion about this result. In our revised manuscript, we improved this sentence as suggested in page 11 (Marked by yellow background, line 373 to 374).

-“However, when we used this drug (calcitriol) to treat embryos at 3 dpf, it was lethal to larval zebrafish”. At which dose? Maybe reducing the dose of calcitrol the Authors could have less lethality and expansion of HSPCs? I would suggest to discuss about it in the revised version of the manuscript.

Thanks for your kind suggestion. In our revised manuscript, we added comments at corresponding part as suggested on page 12 (Marked by yellow background, line 381 to 384)

FIGURES

-Figure 2 legend in situ in italic

Thank you very much! We italicized “in situ” in our revised manuscript accordingly.

-Figure 3 I would suggest to put the control DMSO in the upper part of the figure replacing the scheme of the wells.

Many thanks for your kind suggestion, we reorganized Figure 3A in our revised manuscript as suggested.

-Figure 3A in the legend in situ in italics

Thanks for your kind suggestion, in our revised manuscript, we italicized “in situ” in Figure 3A legend.

-Figure 4G ergocalcipherol in the box. The L is lacking

Many thanks for your valuable advice. We apologize for the spelling mistake. We corrected “ergocalciphero” to “ergocalcipherol” in our revised manuscript.

-Figure 5A is not clear I would suggest to remove it.

Thanks for your kind suggestion. In our revised manuscript, we presented more clear figures to replace previous Figure 5A. Many thanks again for your kind suggestions and comments!